# Magnitude-sensitive preference formation

**Nisheeth Srivastava**[*]
Department of Psychology
University of San Diego
La Jolla, CA 92093
nisheeths@gmail.com

**Edward Vul**
Department of Psychology
University of San Diego
La Jolla, CA 92093
edwardvul@gmail.com

**Paul R Schrater**
Dept of Psychology
University of Minnesota
Minneapolis, MN, 55455
schrater@umn.edu

## Abstract

Our understanding of the neural computations that underlie the ability of animals to choose among options has advanced through a synthesis of computational modeling, brain imaging and behavioral choice experiments. Yet, there remains a gulf between theories of preference learning and accounts of the real, economic choices that humans face in daily life, choices that are usually between some amount of money and an item. In this paper, we develop a theory of magnitude-sensitive preference learning that permits an agent to rationally infer its preferences for items compared with money options of different magnitudes. We show how this theory yields classical and anomalous supply-demand curves and predicts choices for a large panel of risky lotteries. Accurate replications of such phenomena without recourse to utility functions suggest that the theory proposed is both psychologically realistic and econometrically viable.

## 1 Introduction

While value/utility is a useful abstraction for macroeconomic applications, it has little psychological validity [1]. Valuations elicited in laboratory conditions are known to be extremely variable under different elicitation conditions, liable to anchor on arbitrary observations, and extremely sensitive to the set of options presented [2]. This last property constitutes the most straightforward refutation of the existence of object-specific utilities. Consider for example, an experiment conducted by [3], where subjects were endowed with a fixed amount of money, which they could use across multiple trials to buy out of receiving an electric shock of one of three different magnitudes (see left panel in Figure 1). The large systematic differences found in the prices for different shock magnitudes that subjects in this study were willing to pay demonstrate the absence of any fixed psychophysical measurements of value. Thus, while utility maximization is a mathematically useful heuristic in economic applications, it is unlikely that utility functions can represent value in any significant psychological sense.

Neurological studies also demonstrate the existence of neuron populations sensitive not to absolute reward values, but to one of the presented options being better relative to the others, a phenomenon called comparative coding. Comparative coding was first reported in [4], who observed activity in the orbito-frontal neurons of monkeys when offered varying juice rewards presented in pairs within separate trial blocks in patterns that depended only on whether a particular juice is preferred within its trial. Elliott et al. [5] found similar results using fMRI in the medial orbitofrontal cortex of human subjects a brain region known to be involved in value coding. Even more strikingly, Plassmann et al [6] found that falsely assigning a high price to a particular item (wine) caused both greater self-reported experienced pleasantness (EP) (see right panel of Figure 1) and greater mOFC activity indicative of pleasure. What is causing this pleasure? Where is the 'value' assigned to the pricier wine sample coming from?

---

[*]Corresponding author: nisheeths@gmail.com

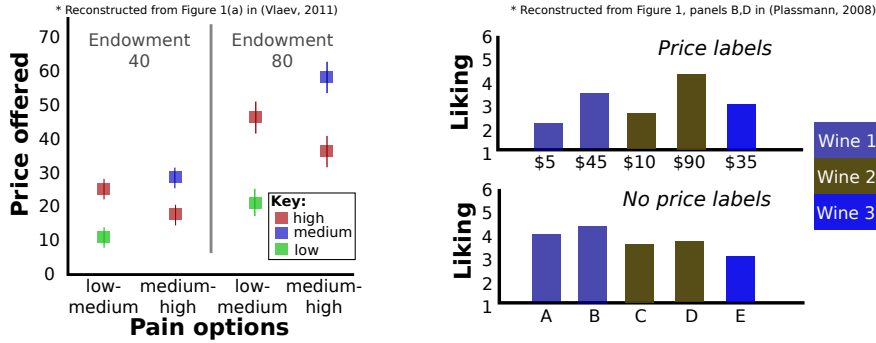

Figure 1: Valuations of options elicited in the lab can be notoriously labile. *Left:* An experiment where subjects had to pay to buy out of receiving electric shock saw subjects losing or gaining value for the price of pain of particular magnitudes both as a function of the amount of money the experimenters initially gave them and the relative magnitude of the pair of shock options they were given experience with. *Right:* Subjects asked to rate five (actually three) wines rated artificially highly-priced samples of wine as more preferable. Not only this, imaging data from orbitofrontal cortex showed that they actually experienced these samples as more pleasurable.

Viewed in light of these various difficulties, making choices for options that involve magnitudes, appears to be a formidable challenge. However humans, and even animals [7] are well-known to perform such operations easily. Therefore, one of two possibilities holds: one, that it is possible, notwithstanding the evidence laid out above, for humans to directly assess value magnitudes (except in corner cases like the ones we describe); two, that some alternative set of computations permits them to behave *as if* they can estimate value magnitudes. This paper formalizes the set of computations that operationalizes this second view.

We build upon a framework of preference learning proposed in [8] that avoids the necessity for assuming psychophysical access to value and develop a model that can form preferences for quantities of objects directly from history of past choices. Since the most common modality of choices involving quantities in the modern world is determining the prices of objects, pricing forms the primary focus of our experiments. Specifically, we derive from our theory (i) classical and anomalous supply-demand curves, and (ii) choice predictions for a large panel of risky lotteries. Hence, in this paper we present a theory of magnitude-sensitive preference formation that, as an important special case, provides an account of how humans learn to value money.

## 2 Learning to value magnitudes

### 2.1 Rational preference formation

Traditional treatments of preference learning (e.g. [9]) assume that there is some hidden state function $U : \mathcal{X} \to \mathbb{R}_+$ such that $x \succ x'$ iff $U(x) > U(x') \, \forall x' \in \mathcal{X}$, where $\mathcal{X}$ is the set of all possible options. Preference learning, in such settings, is reduced to a task of statistically estimating a monotone distortion of U, thereby making two implicit assumptions (i) that there exists some psychophysical apparatus that can compute hedonic utilities and (ii) that there exists some psychophysical apparatus capable of representing absolute magnitudes capable of comparison in the mind. The data we describe above argues against either possibility being true. In order to develop a theory of preference formation that avoids commitments to psychophysical value estimation, a novel approach is needed.

Srivastava & Schrater [8] provide us with the building blocks for such an approach. They propose that the process of learning preferences can be modeled as an ideal Bayesian observer directly learning 'which option among the ones offered is best', retaining memory of which options were presented to it at every choice instance. However, instead of directly remembering option sets, their model allows for the possibility that option set observations map to latent *contexts* in memory. In practice, this mapping is assumed to be identified in all their demonstrations. Formally, the computation corresponding to utility in this framework is $p(r|x, o)$, which is obtained by marginalizing

over the set of latent contexts $\mathcal{C}$,

$$D(x) = p(r|x, o) = \frac{\sum_c^{\mathcal{C}} p(r|x, c)p(x|c)p(c|o)}{\sum_c^{\mathcal{C}} p(x|c)p(c|o)}, \tag{1}$$

where it is understood that the *context* probability $p(c|o) = p(c|\{o_1, o_2, \cdots, o_{t-1}\})$ is a distribution on the set of all possible contexts incrementally inferred from the agent's observation history. Here, $p(r|x, c)$ encodes the probability that the item $x$ was preferred to all other items present in choice instances linked with the context $c$, $p(x|c)$ encodes the probability that the item $x$ was present in choice sets indexed by the context $c$ and $p(c)$ encodes the frequency with which the observer encounters these contexts.

The observer also continually updates $p(c|o)$ via recursive Bayesian estimation,

$$p(c^{(t)}|o^{(1:t)}) = \frac{p(o^{(t)}|c)p(c|o^{(1:t-1)})}{\sum_c^{\mathcal{C}} p(o^{(t)}|c)p(c|o^{(1:t-1)})}, \tag{2}$$

which, in conjunction with the desirability based state preference update, and a simple decision rule (e.g. MAP, softmax) yields a complete decision theory.

While this theory is complete in the formal sense that it can make testable predictions of options chosen in the future given options chosen in the past, it is incomplete in its ability to represent options: it will treat a gamble that pays \$20 with probability 0.1 against safely receiving \$1 and one that pays \$20000 with probability 0.1 against safely receiving \$1 as equivalent, which is clearly unsatisfactory. This is because it considers only simple cases where options have nominal labels. We now augment it to take the information that magnitude *labels*[1] provide into account.

## 2.2 Magnitude-sensitive preference formation

Typically, people will encounter monetary labels $m \in \mathcal{M}$ in a large number of contexts, often entirely outside the purview of the immediate choice to be made. In the theory of [8] incorporating desirability information related to $m$ will involve marginalizing across all these contexts. Since the set of such contexts across a person's entire observation history is larg, explicit marginalization across all contexts would imply explicit marginalization across every observation involving the monetary label $m$, which is unrealistic. Thus information about contexts must be compressed or summarized[2].

We can resolve this by assuming that instead that animals generate contexts as clusters of observations, thereby creating the possibility of learning higher-order abstract relationships between them. Such models of categorization via clustering are widely accepted in cognitive psychology [10].

Now, instead of recalling all possible observations containing $m$, an animal with a set of observation clusters (contexts) would simply sample a subset of these that would be representative of all contexts wherein observations containing $m$ are statistically typical. In such a setting, $p(m|c)$ would correspond to the observation likelihood of the label $m$ being seen in the cluster $c$, $p(c)$ would correspond to the relative frequency of context occurrences, and $p(r|x, m, c)$ would correspond to the inferred value for item $x$ when compared against monetary label $m$ while the active context $c$. The remaining probability term $p(x|m)$ encodes the probability of seeing transactions involving item $x$ and the particular monetary label $m$. We define $r$ to take the value 1 when $x \succ x' \forall x' \in \mathcal{X} - \{x\}$. Following a similar probabilistic calculus as in Equation 1, the inferred value of $x$ becomes $p(r|x)$ and can be calculated as,

$$p(r|x) = \frac{\sum_m^{\mathcal{M}} \sum_{\mathcal{C}} p(r|x, m, c)p(x|m)p(m|c)p(c)}{\sum_m^{\mathcal{M}} \sum_{\mathcal{C}} p(x|m)p(m|c)p(c)}, \tag{3}$$

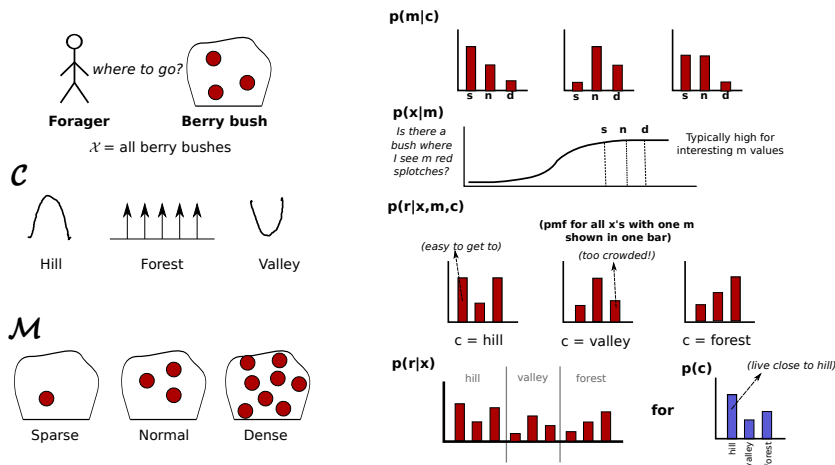

Figure 2: Illustrating a choice problem an animal might face in the wild (left) and how the intermediate probability terms in our proposed model would operationalize different forms of information needed to solve such a problem (right). Marginalizing across situation contexts and magnitude labels tells us what the animal will do.

with the difference from the earlier expression arising from an additional summation over the set $\mathcal{M}$ of monetary labels that the agent has experience with.

Figure 2 illustrates how these computations could be practically instantiated in a general situation involving magnitude-sensitive value inference that animals could face. Our hunter-gatherer ancestor has to choose which berry bush to forage in, and we must infer the choice he will make based on recorded history of his past behavior. The right panel in this figure illustrates natural interpretations for the intermediate conditional probabilities in Equation 3. The term $p(m|c)$ encodes prior understanding of the fertility differential in the soils that characterize each of the three active contexts. The $p(r|x, m, c)$ term records the history of the forager's choice *within* the context in via empirically observed relative frequencies. What drives the forager to prefer a sparsely-laden tree on the hill instead of the densely laden tree in the forest in our example, though, is his calculation of the underlying context probability $p(c)$. In our story, because he lives near the hill, he encounters the bushes on the hill more frequently, and so they dominate his preference judgment. A wide palette of possible behaviors can be similarly interpreted and rationalized within the framework we have outlined.

What exactly is this model telling us though that we aren't putting into it ourselves? The only strong constraint it imposes on the form of preferences currently is that they will exhibit context-specific consistency, viz. an animal that prefers one option over another in a particular context will continue to do so in future trials. While this constraint itself is only valid if we have some way of pinning down particular contexts, it is congruent with results from marketing research that describe the general form of human preferences as being ' arbitrarily coherent' - consumer preferences are labile and sensitive to changes in option sets, framing effects, loss aversion and a host of other treatments but are longitudinally reliable within these treatments [2]. For our model to make more interesting economic predictions, we must further constrain the form of the preferences it can emit to match those seen in typical monetary transactions; we do this by making further assumptions about the intermediate terms in Equation 3 in the next three sections that describe economic applications.

## 3 Living in a world of money

Equation 3 gives us predictions about how people will form preferences for various options that co-occur with money labels. Here we specialize this model to make predictions about the value of options that *are money labels*, viz. fiat currency. The institutional imperatives of legal tender impose a natural ordering on preferences involving monetary quantities. *Ceteris paribus*, subjects will prefer a larger quantity of money to a smaller quantity of money. Thus, while the psychological de-

sirability pointer could assign preferences to monetary labels capriciously (as an infant who prefers the drawings on a \$1 bill to those on a \$100 bill might), to model desirability behavior corresponding to knowledgeable use of currency, we constrain it to follow arithmetic ordering such that,

$$x_{m^*} \succ x_m \Leftrightarrow m^* > m \ \forall m \in \mathcal{M}, \tag{4}$$

where the notation $x_m$ denotes an item (currency token) $x$ associated with the money label $m$. Then, Equation 3 reduces to,

$$p(r|x_{m^*}) = \frac{\sum_m^{\mathcal{M}'} \sum_{\mathcal{C}} p(x|m)p(m|c)p(c)}{\sum_m^{\mathcal{M}} \sum_{\mathcal{C}} p(x|m)p(m|c)p(c)}, \tag{5}$$

where $\max(\mathcal{M}') \leq m^*$, since the contribution to $p(r|x, m, c)$ for all larger $m$ terms, is set to zero by the arithmetic ordering condition; the $p(x|m)$ term binds $x$ to all the $m's$ it has been seen with before.

Assuming no uncertainty about which currency token goes with which label, $p(x|m)$ becomes a simple delta function pointing to $m$ that the subject has experience with, and Equation 5 can be rewritten as,

$$p(r|x) = \frac{\int_0^{m^*} \sum_{\mathcal{C}} p(x|m, c)p(m|c)p(c)}{\int_0^{\infty} \sum_{\mathcal{C}} p(x|m, c)p(m|c)p(c)}. \tag{6}$$

If we further assume that the model gets to see all possible money labels, i.e. $\mathcal{M} = \mathbb{R}_+$, this can be further simplified as,

$$p(r|x) = \frac{\int_0^{m^*} \sum_{\mathcal{C}} p(m|c)p(c)}{\int_0^{\infty} \sum_{\mathcal{C}} p(m|c)p(c)}, \tag{7}$$

reflecting strong dependence on the shape of $p(m)$, the empirical distribution of monetary outcomes in the world.

What can we say about the shape of the general frequency distribution of numbers in the world? Numbers have historically arisen as ways to quantify, which helps plan resource foraging, consumption and conservation. Scarcity of essential resources naturally makes being able to differentiate small magnitudes important for selection fitness. This motivates the development of number systems where objects counted frequently (essential resources) are counted with small numbers (for better discriminability). Thus, it is reasonable to assume that, in general, larger numbers will be encountered relatively less frequently than smaller ones in natural environments, and hence, that the functions $p(m)$ and $p(c)$ will be monotone decreasing[3]. For analytical tractability, we formalize this assumption by setting $p(m|c)$ to be gamma distributed on the domain of monetary labels, and $p(c)$ to be an exponential distribution on the domain of the typical 'wealth' rate of individual contexts.

The wealth rate is an empirically accessible index for the set of situation contexts, and represents the typical (average) monetary label we expect to see in observations associated with this context. Thus, for instance, the wealth rate for 'steakhouses' will be higher than that of 'fast food'. For any particular value of the wealth rate, the 'price' distribution $p(m|c)$ will reflect the relative frequencies of seeing various monetary labels in the world in observations typical to context $c$. The gamma/log-normal shape of real-world prices in specific contexts is well-attested empirically. The wealth rate distribution $p(c)$ can be always made monotone decreasing simply by shuffling the order of presentation of contexts in the measure of the distribution.

With these distributional assumptions, the marginalized product $p(m)$ is assured to be a Pareto distribution. Data from [12] as well as supporting indirect observations in [13], suggest that we are on relatively safe ground by making such assumptions for the general distribution of monetary units in the world [14]. This set of assumptions further reduces Equation 7 to,

$$p(r|x) = \psi(x_{m^*}), \tag{8}$$

where $\psi(\cdot)$ is the Pareto c.d.f.

Reduced experience with monetary options will be reflected in a reduced membership of $\mathcal{M}$. Sampling at random from $\mathcal{M}$ corresponds to approximating $\psi$ with a limited number of samples. So long as the sampling procedure is not systematically biased away from particular $x$ values, the resulting curve will not be qualitatively different from the true one. Systematic differences will arise, though, if the sampling is biased by, say, the range of values observers are known to encounter. For instance, it is reasonable to assume that the wealth of a person is directly correlated with the upper limit of money values they will see. Substituting this upper limit in Equation 7, we obtain a systematic difference in the curvature of the utility function that subjects with different wealth endowments will have for the same monetary labels. The trend we obtain from a simulation (see gray inset in Figure 3) with three different wealth levels ($1000, $10000 and $ 1 million) matches the empirically documented increase in relative risk aversion (curvature of the utility function) with wealth [15]. Observe that the log concavity of the Pareto c.d.f. has the practical effect of essentially converting our inferred value for money into a classical utility function. Thus, using two assumptions (number ordering and scarcity of essential resources), we have situated economic measurements of preference as a special, fixed case of a more general dynamic process of desirability evaluation.

## 4   Modeling willingness-to-pay

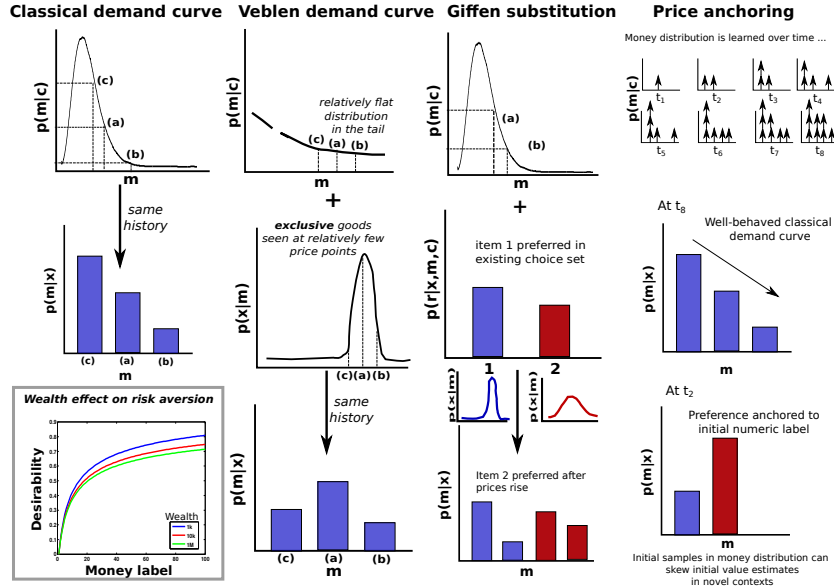

Figure 3: Illustrating derivations of pricing theory predictions for goods of various kinds from our model.

Having studied how our model works for choices between items that all have money labels, the logical next step is to study choices involving one item with a money label and one without, i.e., pricing. Note that asking how much someone values an option, as we did in the section above, is different from asking if they would be willing to buy it at a particular price. The former corresponds to the term $p(r|x)$, as defined above. The latter will correspond to $p(m|r,x)$, with $m$ being the price the subject is willing to pay to complete the transaction. Since the contribution of all terms where $r = 0$, i.e. the transaction is not completed, is identically zero this term can be computed as,

$$p(m|x) = \frac{\sum_{\mathcal{C}} p(x|m)p(m|c)p(c)}{\sum_m^{\mathcal{M}} \sum_{\mathcal{C}} p(x|m)p(m|c)p(c)},\tag{9}$$

further replacing the integral over $\mathcal{M}$ with an integral over the real line as in Equation 5 for analytical tractability when necessary.

What aspects of pricing behavior in the real world can our model explain? Interesting variations in pricing arise from assumptions about the money distribution $p(m|c)$ and/or the price distribution $p(x|m)$. Figure 3 illustrates our model's explanation for three prominent variations of classical

demand curves documented in the microeconomics literature. Consumers typically reduce preference for goods when prices rise, and increases it when prices drop. This fact about the structure of preferences involved in money transactions is replicated in our model (see first column in Figure 3) via the reduction/increase of the contribution of the $p(m|c)$ term to the numerator of Equation 9. Marketing research reports anomalous pricing curves that violate this behavior in some cases. One important case comprises of Veblen goods, wherein the demand for high-priced *exclusive* goods drops when prices are lowered. Our model explains this behavior (see second column in Figure 3) via unfamiliarity with the price reflected in a lower contribution from the price distribution $p(x|m)$ for such low values. Such non-monotonic preference behavior is difficult for utility-based models, but sits comfortably within ours, where familiarity with options at typical price points drives desirability. Another category of anomalous demand curves comes from Giffen goods, which rise in demand upon price increases because another substitute item becomes too expensive. Our approach accounts for this behavior (see third column in Figure 3) under the assumption that price changes affect the Giffen good less because its price distribution has a larger variance, which is in line with empirical reports showing greater price inelasticity of Giffen goods [16].

The last column in Figure 3 addresses an aspect of the temporal dynamics of our model that potentially explains both (i) why behavioral economists can continually find new anchoring results (e.g. [6, 2]) and (ii) why classical economists often consider such results to be marginal and uninteresting [17]. Behavioral scientists running experiments in labs ask subjects to exhibit preferences for which they may not have well-formed price and label distributions, which causes them to anchor and show other forms of preference instability. Economists fail to find similar results in their field studies, because they collect data from subjects operating in contexts for which their price and label distributions are well-formed. Both conclusions fall out of our model of sequential preference learning, where initial samples can bias the posterior, but the long-run distribution remains stable. Parenthetically, this demonstration also renders transparent the mechanisms by which consumers process rapid inflationary episodes, stock price volatility, and transferring between multiple currency bases. In all these cases, empirical observations suggests inertia followed by adaptation, which is precisely what our model would predict.

## 5 Modeling risky monetary choices

Finally, we ask: how well can our model fit the choice behavior of real humans making economic decisions? The simplest economic setup to perform such a test is in predicting choices between risky lotteries, since the human prediction is always treated as a stochastic choice preference that maps directly onto the output of our model. We use a basic expected utility calculation, where the desirability for lottery options is computed as in Equation 8. For a choice between a risky lottery $x_1 = \{m_h, m_l\}$ and a safe choice $x_2 = m_s$, with a win probability $q$ and where $m_h > m_s > m_l$, the value calculation for the risky option will take the form,

$$p(r|x) = \frac{\int_{m_s}^{m_h} p(m|c)p(c)}{\int_0^\infty p(m|c)p(c)}, \text{in wins} \tag{10}$$

$$p(r|x) = \frac{\int_{m_s}^{m_l} p(m|c)p(c)}{\int_0^\infty p(m|c)p(c)}, \text{in losses} \tag{11}$$

$$\Rightarrow EV(risky) = q\left(\psi_x(m_h) - \psi_x(m_s)\right) + (1-q)\left(\psi_x(m_l) - \psi_x(m_s)\right). \tag{12}$$

where $\psi(\cdot)$ is the c.d.f. of the Pareto distribution on monetary labels $m$ and $p(x)$ is the given lottery probability.

Using Equation 12, where $\psi$ is the c.d.f of a Pareto distribution, ($\theta = \{2.9, 0.1, 1\}$ fitted empirically), assuming that subjects distort perceived probabilities [18] via an inverse-S shaped weighting function[4], and using an $\epsilon$-random utility maximization decision rule[5], we obtain choice predictions

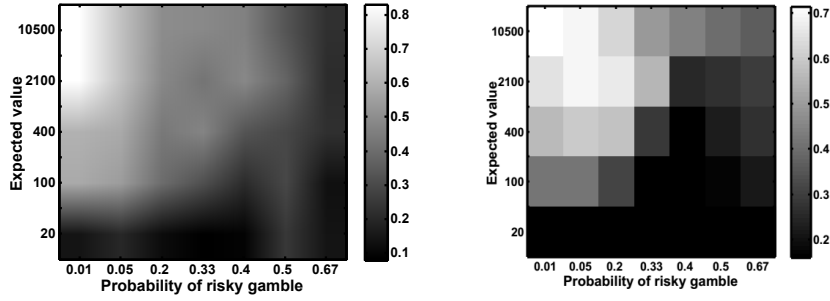

Figure 4: Comparing proportion of subjects selecting risky options predicted by our theory with data obtained in a panel of 35 different risky choice experiments. The x-axis plots the probability of the risky gamble; the y-axis plots the expected value of gambles scaled to the smallest EV gamble. Left: Choice probabilities for risky option plotted for 7 p values and 5 expected value levels. Each of the 35 choice experiments was conducted using between 70-100 subjects. Right: Choice probabilities predicted by relative desirability computing agents in the same 35 choice experiments. Results are compiled by averaging over 1000 artificial agents.

that match human performance (see Figure 4) on a large and comprehensive panel of risky choice experiments obtained from [19] to within statistical confidence[6].

# 6   Conclusion

The idea that preferences about options can be directly determined psychophysically is strongly embedded in traditional computational treatments of human preferences, e.g. reinforcement learning [20]. Considerable evidence, some of which we have discussed, suggests that the brain does not in fact, compute value [3]. In search of a viable alternative, we have demonstrated a variety of behaviors typical of value-based theories using a stochastic latent variable model that simply tracks the frequency with which options are seen to be preferred in latent contexts and then compiles this evidence in a rational Bayesian manner to emit preferences. This proposal, and its success in explaining fundamental economic concepts, situates the computation of value (as it is generally measured) within the range of abilities of neural architectures that can only represent relative frequencies, not absolute magnitudes.

While our demonstrations are computationally simple, they are substantially novel. In fact, computational models explaining any of these effects even in isolation are difficult to find [1]. While the results we demonstrate are preliminary, and while some of the radical implications of our predictions about the effects of choice history on preferences ("you will hesitate in buying a Macbook for $100 because that is an unfamiliar price for it"[7]) remain to be verified, the plain ability to describe these economic concepts within an inductively rational framework without having to invoke a psychophysical value construct by itself constitutes a non-trival success and forms the essential contribution of this work.

**Acknowledgments**

NS and PRS acknowledge funding from the Institute for New Economic Thinking. EV acknowledges funding from NSF CPS Grant #1239323.

---

$\epsilon$. The value of $\epsilon$ is fitted to the data; we used $\epsilon = 0.25$, the value that maximized our fit to the endpoints of the data. Since we are computing risk attitudes over a population, we should ideally also model stochasticity in utility computatation.

[6]While [19] do not give standard deviations for their data, we assume that binary choice probabilities can be modeled by a binomial distribution, which gives us a theoretical estimate for the standard deviation expected in the data. Our optimal fits lie within 1 SD of the raw data for 34 of 35 payoff-probability combinations, yielding a fit in probability.

[7]You will! You'll think there's something wrong with it.

## Footnotes

[1]Note that taking monetary labels into account is not the same as committing to a direct psychophysical evaluation of money. In our account, value judgments are linked not with magnitudes, but with labels, that just happen to correspond to numbers in common practice.

[2]Mechanistic considerations of neurobiology also suggest sparse sampling of prior contexts. The memory and computational burden of recalculating preferences for an ever-increasing $\mathcal{C}$ would quickly prove insuperable.

[3]Convergent evidence may also be found in the Zipfian principle of communication efficiency [11]. While it might appear incongruous to speak of differential efficiency in communicating numbers, recall that the historical origins of numbers involved tally marks and other explicit token-based representations of numbers which imposed increasing resource costs in representing larger numbers.

[4]We use Prelec's version of this function, with the slope parameter $\gamma$ distributed $\mathcal{N}(0.65, 0.2)$ across our agent population. The quantitative values for $\gamma$ are taken from (Zhang & Maloney, 2012).

[5]$\epsilon$-random decision utility maximization is a simple way of introducing stochasticity into the decision rule, and is a common econometric practice when modeling population-level data. It predicts that subjects pick the option with higher computed expected utility with a probability $1 - \epsilon$, and predict randomly with a probability

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
