[Reviews · NeurIPS 2014]

Submitted by Assigned_Reviewer_1

This paper presents a rational model of economic choice, where values are replaced by inferences about relative rank in a context. The model is shown to provide a good fit to pricing data and risky-choice experiments.Overall, I found the paper interesting, but somewhat too narrow and inaccessible for a NIPS audience. There was lots of insufficiently explained economic jargon (e.g. Veblen and Giffen goods), which are key for understanding the contribution. Moreover, the point of the model was hard for me to pull out of the paper. The model was motivated as psychological realistic alternative to utility calculations, but then uses Bayesian inference as a psychological calculation (not as rational model).

Other points:

-Good intro, but a bit too aggressive against the values of utilities. Well-written and fairly clear.

-Variables not obvious in equations. For example, o seems to be observations, but that is never mentioned.

-To reiterate, something not above, I was very confused about the point of this model. It was motivated as psychological realistic alternative to utility calculations, but then uses Bayesian inference as psychological calculation (not as rational model). I don’t see how this improves over the Decision-by-Sampling model, which provides a more realistic psychological grounding and already covers the some of the key features o
What is the contribution? Is it the rational model/analysis?

-Hanging a lot hang on number frequency means there should be all sorts of weird quirks at large, common numbers (e.g., $100 vs $101). Is there any evidence or data that these frequency anomalies exist? And, I don’t see how this guarantees a Pareto distribution? Little support is given to the key claim that number space will follow this distribution, and, intuitively, I don’t see it all. The numbers are certainly not strictly descending in frequency in many contexts. Imagine buying car. $10 is a lot less frequent than $15000. Finally, there is a lot of work on subjective number representation that is not cited here, but also very relevant. Number (as do most senses) follows Weber’s law, whereby larger numbers are less discriminable than smaller numbers (think 1 vs. 2 or 200 vs. 201; see, for example, Dehaene, 2003, in Trends in Cognitive Science).

- Figures 2 and 3 had tones of information, but very little guidance, and I had lots of trouble making sense of them. For example, in Figure 2, what is m (in s, m, d) in top two panels on left? Normal? Not n? And what are the three plots on the topmost panel (presumably hill, valley, and forest, but that’s not labeled). Also, in Fig 2., I like the attempt to explain the pieces of the equation intuitively, but was admittedly baffled by what the bottom 3 panels on the right are supposed to represent. pmf = ? The text only covers some of the figure and is thus unhelpful.

Minor points:

line 080: direct utile estimation = ?
line 082: “from the history of past choices” (missing article)
line 107: identitied?
lines 145: sampling a representative context sounds far from simple
line 151: could be more simply stated in words than the equation, to wit: “when x is the most preferred option in a context”. At the very least, unpack the equation.
line 194. “in via”
line 267. [?]
line 363. Is q or p(x) the lottery probability?
line 377. computatation
line 409-411. A bizarre claim to make about RL and back up with the Sutton & Barto book.

Ref 19 is mangled: The Sutton & Barto book (not Barto & Sutton) was published by MIT press, which is in Cambridge, Mass. (not U of Cambridge).
Summary: This paper presents a rational model of economic choice, where values are replaced by inferences about relative rank in a context. The model is shown to provide a good fit to pricing data and risky-choice experiments. Overall, I found the paper interesting, but somewhat too narrow and inaccessible for a NIPS audience.

Submitted by Assigned_Reviewer_6

This paper introduces a theory of magnitude-sensitive preference learning that does not rely on utility functions, yet replicates classic findings in the field. By not requiring a scalar utility function that assigns value to objects, it becomes a good candidate for psychological plausibility.

The paper is well-written and technically sound. The problem it tackles is ambitious, but the authors manage to present a very simple yet interesting formalism and they do a very good job of showing how it models well-known results in behavioral economics and related fields. I am not an expert in this field to be able to judge how novel this is, but as it stands it's quite convincing, opening potential avenues for experimentation both in behavioral as in neural sciences.

I have a question which is not a criticism of this paper, but makes me curious: would it be possible to account for other phenomena within this model, like the fact that an individual's increased wealth leads to heteroscedasticity in their decisions? That is, in a context where an individual has an endowment of $1M, we expect a greater variance in preferences and decisions agains a context when the endowment is of $10.
Summary: This paper presents an interesting and challenging theory of preference formation based on magnitude that does not rely on a utility function. I believe this paper is most interest in how it opens up a space for new experiments.

Submitted by Assigned_Reviewer_11

I think that many psychologists with Bayesian leanings, who have studied economic value, have intuitively realized the key idea presented in this paper at some point. What makes this paper great, is that the authors have taken an intuitive idea, built on top of Srivastava et al (cite [8]), and added the full force of the Bayesian framework to it. That in it itself makes for a good paper: formalizing an intuitive idea.

Economic theory takes utilities as given i.e. they are magically generated in the agent's head. Some neuroscientists have tried to figure out how values are formed, but most of the literature has looked at accumulators e.g. drift-diffusion models. That leaves a lot of questions unanswered. This paper explains that looking at people as "Bayesian samplers", provides an explain for some behavioural phenomena, such as non-standard demand curves and concave utility for risk.

The writing is very witty, but somewhat irreverent towards economists, which can be typical for outsiders who don't know the history or nuances of the subject. The authors should instead note in the paper that economists always take utilities as given, and are not interested in how they come about. That still makes this paper very interesting from a psychological and cognitive science point of view.

I would conditionally give this paper a 9 (top 15%) iff the authors can provide some economically relevant thought experiments that make unique predictions from their framework. That would help a broader NIPS audience understand the impact of the paper's main contribution. For example, how does this help explain loss aversion? Or a more simple macro situation: how do people adjust to hyper-inflation (e.g. Zimbabwean dollars) and what biases might arise? For risky choices, a consequence of this paper is that subjects who are used to very large monetary amounts (e.g. traders) would have less concave utility and hence lower risk aversion.
Summary: The authors formalize an intuitive idea of how values are formed by building a Bayesian framework on top of [8]: this provides a reasonable explanation for some phenomena in economics such as non-standard demand curves. This is a great paper and the authors should add some thought experiments to make the contributions more understandable for a general NIPS audience.

Submitted by Assigned_Reviewer_45

This paper claims to propose a theory of magnitude sensitive preference learning. The equations in this paper are fairly straight forward. Yet, it was very hard to understand some basic questions were unanswered at the end of reading this paper a few times, like:
1. What is the main purpose of this paper? How is this theory useful in practice?
2. The authors give a few examples in the paper to serve as illustration of their theory, but they are also quite confusing.
3. How is this work relevant to a large machine learning audience at NIPS?

The paper has many cryptic sentences that I could not really understand, e.g:
The institutional imperatives of legal tender impose a natural ordering on preferences involving monetary quantities. Ceteris paribus, subjects will prefer a larger quantity of money to a smaller quanity of money.

e.g. "there is no comparable demonstration of endogenous origin of traditional and anomalous demand curves, wealth effects and arbitrary coherence in the literature"

And some claims like the following which are hard to verify or prove:
Thus, it is reasonable to assume that, in general, larger numbers will be encountered relatively less frequently than smaller ones in natural environments, and hence, that both the functions p(m) and p(m|c) will be decreasing on m3.
Summary: Unfortunately, I am not sure what the main point of this paper is or how the theory proposed is useful. It is quite possible that I did not understand the main point of the paper, but I don't see any strong merits for the paper to be accepted.
Author Feedback
Author rebuttal: All reviewers asked for clearer demonstrations of the theory. Concomitantly answering reviewers’ questions, we present four predictions of the theory, (all incompatible with utility-based accounts),

(i) Hyperinflation (reviewer_11): in lieu of this exotic case, consider a more humdrum analogue - consumer behavior with foreign currencies which are multiples or fractions of their native currency. Our theory predicts a basic anchoring effect: people will underspend when the foreign currency’s face value is a multiple of their native currency’s and overspend when it is a fraction of it. Their existing p(m|c) distributions act as anchors, as illustrated in the rightmost column of Figure 3. We also predict that this effect disappears with increased exposure to the foreign context, again following the dynamics outlined in our theory. Interestingly, this is precisely the pattern of results seen empirically (Raghubir & Srivastava, 2002).

(ii) Are round numbers special? (reviewer_1) Our theory does in fact predict that such numbers will prove to be more desirable price points, and this fact is well-documented in literature on stock market prices and consumer behavior around the world (see e.g. (Osborne,1962),(Sonnemans, 2003)). In fact, technical traders frequently try to use this cognitive effect to their advantage.

(iii) Loss aversion (reviewer_11) is complicated, with aspects that our model cannot explain. We can explain one class of loss aversive behavior, the endowment effect: people require more money to sell something than they are willing to pay for it. Our explanation relies on a sampling approximation of the full Bayesian model. Willingness-to-pay/accept is measured using the highest/lowest accepted transaction value in our model, i.e. the highest m that has p(r|x,m,c) > 0. Inability to recall the highest/lowest values will cause a truncation on the low side for WTP (the highest acceptable option is one of many high options that are being sampled from) and the high side for WTA (vice versa), resulting in an endowment effect. Our model predicts that when someone repeatedly conducts both buy-sell transactions with a good, this disparity will go away (because they learn the range of p(r|x,m,c) better), supporting evidence in (Zhao & Kling, 2001).

(iv) Wealth effect on consumption (reviewer_6). Our theory predicts (i) that someone with a low initial endowment (hence exposure to a smaller range of monetary labels) who wins a lottery will have lower price sensitivity for higher priced goods, hence a greater willingness to pay more for the same consumption bundles, and thus a greater marginal propensity to consume, as typically measured econometrically (supporting evidence in e.g. Blundell, Pistaferri & Preston, 2008).

In general, as reviewer_6 observes, the basic contribution of this theory is that it opens up a new experimental space to quantitatively test the effect of history of observations on economic choices.

While decision-by-sampling acknowledges that learning has a role, an absence of context-specific learning dynamics means that it cannot explain context effects (e.g. similarity, attraction, compromise effects, the monetary base-switching example above), order effects (primacy, recency etc.), while we can.

“motivated as psychological realistic alternative to utility calculations, but then uses Bayesian inference as psychological calculation (not as rational model)” - This is a great point. We are making a greater commitment to Bayesian inference than a simple rational analysis would. This work presupposes that Bayesian calculations can be well approximated by simple and psychologically realistic algorithms (Vul et al, 2014).

p(m|c) always exponential? We assume this for ease of exposition, i.e. the exponential-gamma marginalization assures a Pareto p(m), which lets us describe the utility function for money succinctly. In general, p(m|c) will often have a mode, and the theory sets no restrictions on its shape (see e.g. the p(m|c) plot in the rightmost panel of Figure 3).

Number discriminability We had added a relevant reference (Sun et al, 2012) in footnote 3, but it unfortunately went hanging; we will add it in the final version.

Empirical evidence for approximately Pareto p(m) is well-reviewed in (Stewart, Chater & Brown, 2006).

reviewer_1 pointed out a number of typos, which we will fix. Important among them,

(i) the ‘m’ in ‘s m d’ in Figure 2 is a (well-spotted) typo. It should be ‘n’ for normal.
(ii) The plot at the bottom of Figure 2 labeled p(r|x) is plotting the relative desirability of 9 classes of bushes : {sparse, normal, dense} at locations {hill, forest, valley}. We will express this more clearly in the text.
(iii) ‘utile’ references Bentham’s historical unit for psychophysical value.
(iv) q is the lottery probability.
(v) RL cite mangled. Yes, we’ll fix this.

Blundell, R., Pistaferri, L., & Preston, I. (2008). Consumption inequality and partial insurance. The American Economic Review, 1887-1921.

Osborne, M. F. M. (1962). Periodic structure in the Brownian motion of stock prices. Operations Research, 10(3), 345-379.

Raghubir, P., & Srivastava, J. (2002). Effect of face value on product valuation in foreign currencies. Journal of Consumer Research, 29(3), 335-347.

Sonnemans, J. (2003). Price Clustering and Natural Resistance Points in the Dutch Stock Market (No. 03-043/1). Tinbergen Institute.
Sun, J. Z., Wang, G. I., Goyal, V. K., & Varshney, L. R. (2012). A framework for Bayesian optimality of psychophysical laws. Journal of Mathematical Psychology, 56(6), 495-501.

Vul, E., Goodman, N., Griffiths, T. L., & Tenenbaum, J. B. (2014). One and done? optimal decisions from very few samples. Cognitive science, 38(4), 599-637.

Zhao, J., & Kling, C. L. (2001). A new explanation for the WTP/WTA disparity. Economics Letters, 73(3), 293-300.